# Consensus on maturity-related injury risks and prevention in youth soccer: A Delphi study

Joseph Sullivan[1]*, Simon Roberts[1]*, Kevin Enright[1], Martin Littlewood[1], David Johnson[2,3], David Hartley[4]

1 School of Sport and Exercise Sciences, Liverpool John Moore's University, Liverpool, United Kingdom, 2 West Ham United Football Club, London, United Kingdom, 3 Department for Health, University of Bath, Bath, United Kingdom, 4 Celtic Football Club, Glasgow, United Kingdom

* J.F.Sullivan@2022.ljmu.ac.uk (JS); S.Roberts2@ljmu.ac.uk (SR)

**Data Availability Statement:** All relevant data are within the manuscript and its Supporting Information files.

## Abstract

The aim of this study was to achieve consensus from leading sport and exercise science practitioners in professional soccer academies surrounding (i) motivations for maturity-related data collection (ii) maturity-related injury risk factors (iii) how maturity-related data informs injury prevention practices and (iv) the use of bio-banding as an alternative injury prevention strategy. The study adopted an iterative three round online Delphi method, where a series of statements were rated by expert panellists. Consensus agreement was set at ≥70% for all statements. Nine panellists participated in all three rounds (69% response rate). Consensus was achieved for a total of sixteen statements. Panellists agreed that the period during and 12-months post peak height velocity, muscle strength/flexibility imbalances and maturity status (% predicted adult height) as the most important maturity-related injury risk factors. Panellists also agreed that maturity-related data collection is important for injury prevention as well as physical and performance-related purposes, but not for recruitment or retain/release purposes. It was also evident that variability and misunderstanding of key language terms used within the growth and maturation literature exists. It was agreed that practitioners who are responsible for conducting maturational assessments require additional training/education to enhance their application, delivery and outcomes. The findings indicate that maturity-related data collection is part of a multidisciplinary process, dedicated towards the long-term development of players. Greater training and education are required along with increased dissemination of research findings surrounding the full uses for bio-banding. This study provides guidance on maturity-related injury risks and prevention in youth soccer for practitioners.

## Introduction

Performance staff employed in youth soccer academies play a vital role in the development of soccer players by providing physical, psychological, perceptual-cognitive, and sociocultural

**Funding:** The author(s) received no specific funding for this work.

**Competing interests:** The authors have declared that no competing interests exist.

interactions [1]. Youth academy players (i.e., 9–18 years) on talent development pathways may encounter unique physical development challenges due to the individual timing and rate of their biological maturation [2]. For example, youth players of the same chronological age group can vary in biological maturity by as much as 5–6 years [3]. The variability in the timing of the adolescent growth spurt (occurring between 13–15 years in boys) can offer additional complexities for performance staff who wish to implement injury risk and training load management strategies [1, 4, 5].

Some maturity-related injury risk factors that have to be considered by practitioners when implementing injury prevention strategies include changes in body mass index $> 0.3$ kg/m$^2$ per month and accelerated monthly growth rates ($> 0.6$ cm) in stature and the lower-extremities [6], the 12-month time period around peak height velocity [7] (PHV; the fastest rate of growth in stature during the adolescent growth spurt) [8], maturity status (percentage of predicted adult height $< 88\%$ to $> 95\%$) [9], 'adolescent awkwardness' [10], peak weight velocity [5, 11] and training load volume [2]. Quadriceps flexibility $\geq 35°$ and gastrocnemius flexibility $< 0°$ are also associated with the development of apophysitis conditions (e.g., Osgood-Schlatter, Sever's disease) in youth soccer players [12].

Despite previous investigations around maturity-related injury risk factors and monitoring practices [4, 13], it is currently unclear which of the proposed risk factors from previous literature are truly considered as a risk factor and a priority from a sport science practitioner perspective [1]. With this in mind, injury aetiology and prevention models have been proposed to better understand the relationship between injury risk factors and maturity status [14]. For instance, recent work has reported that youth soccer practitioners have a variety of non-invasive (i.e., predictive equations) methods available to use at their disposal, to assist with the prediction of a player's maturity status (the level of maturity at a given time point) [4] and to help determine individual player injury risk via their stage and timing of maturation (the timing and tempo of progress towards an adult biological state) [4, 8]. Regular assessments of growth (changes in stature and limb length that follow the onset of puberty) [4] and maturation is crucial, given that well documented associations between a player's stage of maturation and injury risk/severity exist [15, 16], particularly around the time of PHV. Consequently, the longitudinal assessment of player maturity status and growth offers a method of injury prevention, by profiling 'at risk' players in the academy system, to optimise their physical development [2, 4, 17].

Previous work has confirmed that maturity offset methods such as Mirwald [18], Fransen [19], Moore [20] and percentage of predicted adult height methods such as Khamis-Roche [21] are frequently used by academies for maturational assessments [22], to measure somatic maturity (the degree of growth in overall stature, or of specific dimensions of the body) [23], as opposed to skeletal maturity (the degree of maturation according to the development of skeletal tissue) [23]. However, a recent review reported that no methods produce equivalent estimations of adult height, skeletal age or age at PHV. For example, there were discrepancies between actual and predicted values of adult height (-0.45 to -2.1cm) and age at PHV (0.3–0.75 years) [24]. Moreover, only a moderate agreement (44–50%) was reported for the maturity status classification of players using maturity offset [19] and predicted adult height methods [21, 24] and between non-invasive (i.e., predictive equations) and invasive (i.e., medical imaging) methods (55–68%) [24]. These findings suggest that the non-invasive methods used to assess maturity status and timing in youth players require further validation [10], as this could lead to further implications for training load prescription, the correct identification of the timing/period of PHV and the maturity status classification of players [10]. Further investigation is needed to explore the reasons behind the continued use of these non-invasive methods, given their questionable reliability, with associated error rates for non-invasive methods varying between 1–3% in boys and girls [10] alongside the potential to over/underestimate the timing

of PHV in early and late maturing players respectively [10]. This can prove critical, particularly for players around the age of PHV (13–15 years), where injury incidence within soccer academies is at its peak, coupled with the additional implications of incorrect maturity categorisation on training load and injury risk management [10, 25].

The concept of 'bio-banding' (a method of grouping players together based on maturity status)–[26], has increased in popularity within youth soccer, to reduce the over-selection of early maturing players within academy systems and to technically/tactically challenge early/late maturing players [26]. However, recent studies go a stage further and suggest that it can be used as a method of maturity-related injury prevention [22]. The precise mechanisms to support this claim are currently unknown, and in the absence of any longitudinal randomised control studies or meta-analyses surrounding this concept, further research and dissemination of its findings is needed to understand its full application in practice [27].

At present, limited data exists describing the process of how sports scientists embedded in professional soccer environments collect, interpret and communicate maturity-related data to their colleagues [13]. Furthermore, it is not clearly understood how practitioners use this data to inform decision-making surrounding injury management and player selection strategies [17]. This information can help to bridge the gap between research and practice, by facilitating an understanding of the data analysis and communication strategies used to inform decision-making within academy environments [27]. Given the amount of heterogeneity within this research area, Delphi studies and expert consensus statements can be a useful mechanism for generating new knowledge and transferring the best available research evidence into practice [28], which can create a better understanding of the growth and maturation practices that occur within soccer academies. In the absence of any well-established meta-analyses and randomised control trials within this research area, it is plausible that the Delphi technique could help to guide the way for a homogenous approach within this research area [24] and have a meaningful impact on current injury prevention and maturity-related data collection practices within academy soccer environments, which has been the case in other sports [29].

With this in mind, the aim of the research is to implement a Delphi poll to gain a consensus on the following questions, to bridge the gap between research and practice: (i) What are the primary motivations for capturing maturity-related data in professional soccer club academies in the United Kingdom (UK)? (ii) Which maturity-related injury risk factors are highly considered for prevention among professionals in soccer academies? (iii) How is maturity-related data used in practice to inform injury prevention strategies within professional soccer club academy systems? (iv) What is the perceived role and effectiveness of 'bio-banding' in maturity-related injury prevention among professionals in soccer academies?

## Methods

### Research design

This study adopted a web-based Delphi approach [30], to establish consensus surrounding the importance of maturity-related injury risk factors, data collection techniques and prevention strategies in youth soccer academies in the UK. The Delphi protocol was designed by the research team which included (i) a registered orthopaedic physiotherapy assistant working in both clinical and professional soccer environments (ii) An applied physiologist (PhD) with expertise and published research in soccer related injury risk factors (iii) A performance psychologist working in the English Premier League (iv) An experienced academic with expertise in Delphi procedures. For transparency, the final Delphi protocol was registered on the Open Science Framework (osf.io/57g3f).

## Delphi design

The Delphi process incorporated an iterative series of three online rounds which has been used previously [31]. Typically, the Delphi technique incorporates three rounds of surveys to achieve consensus on a certain topic or issue [32], however if required, more rounds may be included. Consensus is typically achieved when $\geq 70\%$ of panellists agree on a certain response or statement for a given topic [33], and this threshold was applied in the present study. According to Hasson *et al.*, [32] previous Delphi studies have varied in sample sizes between 15–60 panellists, with known issues surrounding data handling and analysis associated with larger sample sizes. Based on previous studies, it was decided that the sample size for the current Delphi poll would be between 11–20 panellists [32]. Previous work has suggested that Delphi studies are effective in research areas where there is limited or contradictory evidence [33]. After consultation, it was decided that a Delphi approach would benefit this research area, given the amount of heterogeneity that is evident within existing literature, resulting from different outcome variables, populations and research designs [24]. It was agreed that using the Delphi technique to gain consensus on emerging topics could help to guide the way for future research in this area.

## Participants

A key consideration for Delphi studies is the identification and inclusion of expert panellists [34]. Using a combination of purposeful and snowball sampling procedures, the research team identified practitioners working in leadership roles in male soccer academy environments using the following job titles: "*Academy Technical Director*", "*Lead Academy Sport Scientist*", "*Academy Head of Sports Science and Medicine*", "*Head of Sports Science and Athletic Development*", "*Head of Academy Performance Support*", "*Head of Medical*", "*Academy Head of Physical Performance*". As well as holding the pre-requisite job title, to be included in the panel, panellists were also required to possess one or more of the following criteria (i) hold a postgraduate qualification (i.e., MSc/MRes/MPhil) or doctorate level qualification (e.g., PhD or Professional Doctorate) in a sport science related discipline (ii) working in a professional soccer academy in the UK with responsibility for collecting maturity-related data (iii) published scientific research in the field of growth and maturation in youth sports. Panellist recruitment was completed between 1st October - 1st November 2023.

Twenty-three industry experts responded to our initial email to participate in the study, however, only thirteen experts agreed to participate in round one. The included panellists' job titles included: Head of Academy Sport Science (N = 6), Lead Academy Sports Scientist (N = 4), Academy Head of Physical Development or Performance (N = 2) and a former Head of Academy Performance (N = 1). Panellists had a range of experience within their job roles, varying from a low of 3-months to a high of 13-years. Specifically, our panel comprised of four panellists working for different English Premier League clubs. Five panellists were currently or previously (last 12 months) employed within English Football League (EFL) Championship clubs. Three panellists worked for different Scottish Premier League clubs and one panellist worked for a Scottish League One club.

## Ethics

Ethical approval for the study was granted on 10/07/2023, by the Liverpool John Moore's University Research Ethics Committee (UREC reference: 23/SPS/036). Written consent was obtained via consent forms sent by email to all panellists who wished to take part in the study. On receipt of the signed consent forms, panellists were advised they were free withdraw from

the study at any stage in the process. The study was conducted in accordance with the principles expressed in the Declaration of Helsinki.

## Methodology

The study protocol and confidentiality statements were forwarded to all panellists via email. The panellists were also provided with a unique username, password and a personalised URL link to complete each Delphi questionnaire. This ensured that each panellist remained anonymous from each other but were known to the lead researcher when each Delphi round was completed. Each survey was developed using specialist JISC online survey software (https://beta.jisc.ac.uk/online-surveys) and all panellists were afforded a maximum of four weeks to complete each round of the Delphi. Prior to round one, the web-based survey was beta-tested by a group of nine postgraduate (i.e., MSc) students, however, no adjustments were required, and no technical issues were reported.

## Round one

The first Delphi survey was divided into three categories based on a previous systematic review, blinded for review. The first category (attitudes toward the reliability of maturity-related data collection methods), contained nine questions with '*Yes*', '*No*' or '*Not sure*' responses, but with the option to include an open free-text response. The aim of this category was to establish the panellists' attitudes and opinions towards maturity-related data collection [13], as well as its impact on injury prevention within youth academy players [1]. The second category (perceptions of important maturity-related injury risk factors for mitigation) required each panellist to rank order a list of eleven maturity-related injury risk factors in relation to their perceived importance for injury prevention strategies [4]. Each proposed risk factor was ranked on a 10-point scale (1 = least important, 10 = most important). The third category (attitudes toward injury prevention practices, policies and data collection methods used at academy clubs), was a series of ten open-ended questions which aimed to establish the efficacy of the current methods and policies used within youth academy settings for the prevention of maturity-related injuries [35].

## Data analysis round one

Responses from each panellist were exported from the JISC survey software to Microsoft Excel for analysis. For the multiple-choice questions, group cumulative frequencies (%) were calculated for each question to determine the level of agreement. For the ranking questions, the mean, median and interquartile range were calculated from the group responses to each question and were presented in the form of a box plot. Prior to round two, each panellist was provided with a breakdown of their individual scores, as well as the distribution of scores across the group. A list of all the anonymous responses to the open-ended questions was also provided. The research team had planned to simply retain items with good levels of agreement, but based on the comments made by panellists, the research team decided to go further by removing, combining and rewording many items into a series of statements. Following analysis of the first-round responses, eighteen statements were created.

## Round two and analysis

A second-round survey which contained the eighteen statements were emailed to each panellist via the web-based platform JISC. Panellists were asked to rank each statement on a 10-point Likert-scale (1 = strongly disagree, 10 = strongly agree), for validity purposes. Those

who agreed that a statement was relevant, but disagreed on the wording were invited to suggest alternatives via an open text response. Panellists were also asked to suggest additional topic areas and items that they felt were important but were not included in the initial survey. Each of the responses were collated, and the numerical rankings were entered onto a Microsoft Excel spreadsheet. The mean, median and interquartile range for each response was calculated. Statements that scored low on relevance were omitted for the subsequent round.

### Round three and analysis

For the final round, the research team analysed all remaining statements that didn't achieve consensus during the previous round. Statements that were rated as neutral (median score = 5–6) were re-worded and were emailed back to panellists in round three to rate again. Two statements were not distributed during round three, as the research team believed that these statements placed a requirement on panellists to have an extensive knowledge around the application of these specific methods and their respective limitations. It was apparent that some professional clubs may not use these methods for assessing maturity status and timing in their youth players and therefore it was deemed inappropriate to score the statement again. One statement was generated based on the responses from panellists in the previous round, giving a total of four statements that were emailed back to all panellists to achieve consensus ($\geq$ 70%). For these statements, panellists were asked to rank their level of agreement for each statement on a 10-point Likert-scale (1 = strongly disagree, 10 = strongly agree), for validity purposes. Those who agreed that a statement was relevant but disagreed on the wording were invited to suggest alternatives via an open-text response.

For all statements that achieved consensus in the previous round, the research team made a conscious effort to improve the wording of these statements based on the comments made by panellists. For these statements, panellists were asked if they were satisfied with the amended statement via a '*Yes*' or '*No*' response. Panellists who remained unsatisfied with the newly worded statement were asked to suggest alternatives via an open-text response. Each of the responses were collated and entered onto a Microsoft Excel spreadsheet. The mean, median and interquartile range for each response was calculated for the four statements that were re-sent to gather a consensus.

## Results

Ten panellists took part in the first two rounds of the Delphi poll (response rate = 77%). One panellist dropped out during round three. Three panellists dropped out before the start of round one and were excluded from analysis. Nine panellists took part in all three rounds (response rate = 69%).

### Round one

There was a consensus (100%) that the regular collection of maturity-related data can aid with injury prevention and facilitate better long-term outcomes regarding player selection and development. There was also a large agreement (70%) that predictive equations for assessing the maturational status and timing of youth players are sub-optimal and require improvement.

For maturity-related injury risk factors, there was a perceived higher importance (median score $\geq$ 7) for accelerated growth rates, muscle strength/flexibility imbalances, abnormal movement mechanics, the period during and after (i.e., 12 months) peak height velocity, previous injury history and a player's maturity status as a percentage of predicted adult height. The least important maturity-related injury risk factors (median score $\leq$ 5) were group maturity status, fluctuations in lean body mass and the period before (i.e., 12 months) PHV. For a full

summary of the results for round one, see S1 File (round one synthesis) and S2 File (round two background report).

## Round two

In round two, eighteen statements were proposed to panellists and consensus (median score = $\geq 7/10$) was achieved on thirteen statements (72%). The statements that achieved consensus are listed below in Table 1 below. For a full summary of the results for round two, see S3 File (round two synthesis) and S4 File (round three background report).

## Round three

For a full summary of the results for round three, see S5 File (round three synthesis). Three statements were re-distributed during round three to achieve consensus. Furthermore, one additional statement was also generated following comments made in the previous round. These additional four statements are listed below:

> "*Growth and maturity data can inform decisions around player selection/deselection, recruitment and profiling for positional requirements until the player is aged 16–18 years.*"

> "*Additional training and education is required surrounding the prescription of interventions for academy players with growth-related conditions such as Severs disease or Osgood-Schlatter's.*"

**Table 1. Statements (N = 13) that achieved consensus in round two.**

| Statement | Median score |
|---|---|
| Reasons for the collection of maturity-related data include concerns about overuse/growth related injuries and to identify players at immediate risk of injury. | 7 |
| Players with deficits in movement efficiency are at greater risk of growth-related injuries. | 7 |
| We have only limited ability to predict which players with deficits in movement efficiency will go on to experience poorer long-term injury risk outcomes. | 7 |
| Functional assessments that explore "adolescent awkwardness" seem a promising approach. In principle, it might help performance staff understand the mechanisms by which deficits in movement competency around PHV increases injury risk. | 7 |
| Maturity-related data allows performance staff to monitor and adjust training load especially for those players closer to PHV. | 7 |
| Growth-related data can be used to identify both early and late maturing players and determine whether players need to play 'up' or 'down' an age group. | 7 |
| Maturity-related data needs to be presented in a manner that coaches will understand, due to the consequences of data misinterpretation on player development. | 10 |
| Medical scanning techniques could provide greater reliability, validity and sensitivity for maturity-related assessment, but non-invasive methods can provide complimentary information. | 7 |
| Players who are before or during PHV would benefit from an increased frequency of maturity and injury screening assessments from 12-week to 6-week intervals. | 7 |
| Longitudinal maturity-related data collection is preferable as it allows for a more accurate assessment of maturation and its effects on injury risk over the course of the season(s). | 7 |
| Accelerated growth rates, imbalances between muscular strength and flexibility, abnormal movement mechanics, the period during and after age at PHV and a players maturity status as a percentage of adult height are the highest priority maturity-related injury risk factors. | 7 |
| Training load management and S&C interventions are the most effective strategies to limit the effect of maturity-related injury risk factors. | 7 |
| Better understanding of the full application of bio-banding and its wider uses are needed for performance staff. | 10 |

"*Performance/sports science staff in academy environments have sufficient knowledge and expertise of taking growth-related measurements and using common maturity assessment methods in practice [e.g. Khamis-Roche., 1994; Mirwald., 2002] to determine a players' maturity status and the timing of the adolescent growth spurt.*"

"*Apophysitis conditions around the hip are more difficult to diagnose than apophysitis conditions around the foot and ankle and require a specialist assessment.*"

Consensus (median score = ≥ 7/10) was achieved for sixteen statements proposed in round three (100%). One statement remained neutral (median score = 5) during round two and three and was subsequently removed from the analysis due to the failure to reach a consensus:

"*Growth and maturity data can inform decisions around player selection/deselection, recruitment and profiling for positional requirements until the player is aged 16–18 years*".

The final list of statements that achieved consensus are presented in Table 2 below:

## Narrative synthesis of consensus statements

In this section, the final statements are presented alongside supplementary evidence provided by panellists to support their reasoning. This section also includes supporting evidence from references where appropriate.

**Statement 1:** *Reasons for the collection of maturity-related data include concerns about overuse growth related injuries and to identify players at immediate or future risk of injury.*

## Supplementary information

In round two, panellists had concerns surrounding the use of the word "*immediate*". This has connotations towards more traumatic mechanisms of injury, which panellists argued are hard to account for with the regular collection of maturity-related data. They commented that the collection of maturity-related data is catered more towards the prevention of overuse and future injuries, caused by repeated and chronic high training loads and volume. Therefore, the statement was amended to include overuse-type injuries with consideration of future injury risk for players pre or circa-PHV [16].

**Statement 2:** *Players with deficits in movement efficiency might demonstrate a greater risk of growth-related injuries, however more research is needed given the quality of current evidence.*

## Supplementary information

'Movement efficiency' is a term that is becoming increasingly common in growth and maturation literature [35]. However, although panellists agreed on the inclusion of the term, there is a lack of consensus surrounding a specific definition for this phrase as stated by one panellist: "*we have a poor understanding of movement efficiency, even here you don't define it.*" (Panellist 1). Panellists seemed to be familiar with the term and its associated features (e.g., adolescent awkwardness/clumsiness, reduced motor control, lower extremity growth) [35], however there were concerns that the current level of evidence to support this claim was low. This was important when re-wording the statement, to appreciate the concerns regarding low quality evidence, given that practitioners seem to rely on experience to discuss this topic.

**Statement 3:** *It is difficult to predict which players with deficits in movement efficiency will go on to experience poorer long-term injury risk outcomes. This could be improved with better equipment and education.*

**Table 2. Statements (N = 16) that achieved consensus (median score = ≥ 7/10) in round two and three.**

| Statement | Median Score |
|---|---|
| Reasons for the collection of maturity-related data include concerns about overuse growth related injuries and to identify players at immediate or future risk of injury. | 7 |
| Players with deficits in movement efficiency might demonstrate a greater risk of growth-related injuries, however more research is needed given the quality of current evidence. | 7 |
| It is difficult to predict which players with deficits in movement efficiency will go on to experience poorer long-term injury risk outcomes. This could be improved with better equipment and education. | 7 |
| Functional assessments that explore "adolescent awkwardness" seem a promising but under investigated approach. In principle, it may facilitate conversations with performance staff to help them understand the mechanisms by which deficits in movement competency around PHV increases injury risk and can subsequently influence on-pitch performance and injury incidence. | 7 |
| Maturity-related data, that is communicated in a timely manner, allows performance staff to monitor and adjust training load especially for those players closer to PHV. However, it should be conducted in a way that considers the individual, their environmental context and any extra-curricular or school activities the individual may partake in. | 7 |
| Growth-related data can be complimented with performance-related data to identify both early and late maturing players and also to determine whether players need to play across younger or older chronological age groups. | 7 |
| Maturity-related data needs to be presented to coaches using a personalised approach based on their individual preferences, due to the consequences of data misinterpretation on player development, selection and training load management. | 10 |
| Medical scanning techniques can provide greater reliability, validity and sensitivity for maturity-related assessments, but are unlikely to be used in a real world setting due to ethical and financial implications. | 7 |
| Players who are before or during PHV, would benefit from an increased frequency of maturity and injury screening assessments from 12-week to 6-week intervals. This could help to closely monitor the physiological processes associated with an increased risk of injury, providing that measurements are taken accurately. | 7 |
| Longitudinal and standardised maturity-related data collection is preferable as it allows for a more accurate representation of maturation and its effects on injury risk over the course of the season(s), as well as identifying other inherent injury risk factors and players who are at an increased risk of injury. | 7 |
| Maturity-related risk factors with the highest consideration for injury prevention include accelerated growth rates, imbalances between muscular strength and flexibility, abnormal movement mechanics, the period during and after age at PHV, reductions in neuromuscular control and a players' maturity status (% predicted adult height). | 7 |
| Multidisciplinary approaches towards training/game load management, S&C interventions and consideration of injury history are the most effective strategies to limit the effect of maturity-related injury risk factors. | 7 |
| It is unrealistic for practitioners to use bio-banding as a method to reduce injury risk without greater training and research dissemination. | 10 |
| Additional training and education is required surrounding the prescription of interventions for academy players with growth-related conditions such as Severs disease or Osgood-Schlatter's. | 10 |
| Qualified performance/sports science staff in academy environments have sufficient knowledge and expertise of taking growth-related measurements and using common maturity assessment methods in practice [e.g. Khamis-Roche, 1994; Mirwald, 2002] to determine a players' maturity status and the timing of the adolescent growth spurt. | 7 |
| Apophysitis conditions around the hip are more difficult to diagnose than apophysitis conditions around the foot and ankle and require a specialist assessment. | 7 |

## Supplementary information

Panellists reported it was difficult to objectively assess 'movement efficiency' given the absence of a clear definition of the term and a lack of valid movement assessment tools feasible for use in real-world settings [36]. Panellists argued that the facilities they have at their club do not

enable a thorough assessment of movement efficiency and equipment availability can have an influence on these types of assessments [35]. It was also apparent that greater education and dissemination of this term is required, to create a homogenous definition and to devise viable methods to assess this concept. This led to consensus once better equipment and education were accounted for in this statement.

**Statement 4:** *Functional assessments that explore "adolescent awkwardness" seem a promising but under investigated approach. In principle, it may facilitate conversations with performance staff to help them understand the mechanisms by which deficits in movement competency around PHV increases injury risk and can subsequently influence on-pitch performance and injury incidence.*

## Supplementary information

Similar to movement efficiency, there is a lack of conceptual clarity around the term 'adolescent awkwardness' [37]. Panellists suggested that this is currently an under investigated area with poor evidence. Some panellists stated that it was a promising approach to supplement performance-related data, in order to hold conversations with coaches around the long-term development of individual players. It should be noted that good performance in functional assessment tests does not necessarily translate into on-pitch performance, therefore further research and dissemination surrounding this concept and how it influences injury risk and on-pitch performance is required. In practice however, there is a lack of standardised measures for assessing 'adolescent awkwardness' [37].

**Statement 5:** *Maturity-related data, that is communicated in a timely manner, allows performance staff to monitor and adjust training load especially for those players closer to PHV. However, it should be conducted in a way that considers the individual, their environmental context and any extra-curricular or school activities the individual may partake in.*

## Supplementary information

This statement clarifies that youth players who are on dual-career pathways (i.e., still in formal education) may participate in extra-curricular activities, which is often encouraged by clubs to avoid early specialisation and to develop transferrable sporting behaviours [38]. Extra-curricular activities can also influence training load that each youth player is exposed to, therefore these activities must be considered when implementing training load management strategies especially for players around the point of PHV.

**Statement 6:** *Growth-related data can be complemented with performance-related data to identify both early and late maturing players and also to determine whether players need to play across younger or older chronological age groups.*

## Supplementary information

This statement confirms that that growth-related data should not be used in isolation to inform decisions around player development and should be complimented with technical and performance-related data to inform these decisions [13]. Panellists were reluctant to use phrases such as "*playing up or playing down*" (Panellist 4) and agreed to the term "*playing across*" (Panellist 7) various age groups, implying that the academy system should be considered as a continuum for development as opposed to isolated age groups. In round three, consensus was established when the wording was changed to acknowledge the combination of maturity and performance-related data to inform decision making, with subtle changes to 'playing across' the age groups rather than simply 'up' or 'down'.

**Statement 7:** *Maturity-related data needs to be presented to coaches using a personalised approach based on their individual preferences, due to the consequences of data misinterpretation on player development, selection and training load management.*

## Supplementary information

Panellists were all in agreement that the way data is presented and visualised has huge implications for stakeholder buy-in and to ensure the various needs of stakeholders are met without ambiguity [39]. The general consensus was that if data is presented and visualised using commonly accepted software (e.g., Tableau, Power BI), this can facilitate with the development of appropriate actions plans to address the issues that are presented in multidisciplinary team meetings [40]. Panellists also pointed out the negative implications and lack of understanding that can emerge from poor data presentation across key stakeholder groups (i.e., performance staff and coaches) [41].

**Statement 8:** *Medical scanning techniques can provide greater reliability, validity and sensitivity for maturity-related assessments, but are unlikely to be used in a real world setting due to ethical and financial implications.*

## Supplementary information

Panellists were aware that invasive methods such as medical scanning provide greater reliability for assessing biological maturation. However, it was also argued that these methods are not always available to clubs, given the issues surrounding cost and ethical considerations of repeated exposures to radiation for youth players [42]. This statement was therefore re-worded to account for the logistical issues associated with invasive methods to achieve consensus. These ethical and financial concerns may offer an explanation for the preference of soccer academies to use non-invasive over invasive methods during maturity assessments.

**Statement 9:** *Players who are before or during PHV, would benefit from an increased frequency of maturity and injury screening assessments from 12-week to 6-week intervals. This could help to closely monitor the physiological processes associated with an increased risk of injury, providing that measurements are taken accurately.*

## Supplementary information

There was an acknowledgment that players suspected of being immediately pre-PHV or circa-PHV would benefit from increased screening from a maturity monitoring and injury perspective [35]. However, panellists re-iterated the importance of accurate data collection protocols, which may not always be the case in academy environments if untrained personnel undertake this role. This was overlooked in the initial statement but was included in the re-wording of the statement to achieve consensus. It was also accepted that longitudinal growth patterns within the maturation process can be identified with an increased frequency of assessments [43].

**Statement 10:** *Longitudinal and standardised maturity-related data collection is preferable as it allows for a more accurate representation of maturation and its effects on injury risk over the course of the season(s), as well as identifying other inherent injury risk factors and players who are at an increased risk of injury.*

## Supplementary information

It was generally accepted that longitudinal monitoring is preferable, to gain a more accurate depiction of maturation and growth on injury risk [43]. This statement required a minimal

amendment to include individual player risk as well as playing group injury risk over multiple seasons.

**Statement 11:** *Maturity-related risk factors with the highest consideration for injury prevention include accelerated growth rates, imbalances between muscular strength and flexibility, abnormal movement mechanics, the period during and after age at PHV, reductions in neuromuscular control and a players' maturity status (% predicted adult height).*

## Supplementary information

These risk factors were combined following multiple responses from round one. Once the statement was agreed upon, there was general consensus over the wording and no further amendments were required.

**Statement 12:** *Multidisciplinary approaches towards training/game load management, S&C interventions and consideration of injury history are the most effective strategies to limit the effect of maturity-related injury risk factors.*

## Supplementary information

The initial statement simply stated "*training load management and strength and conditioning interventions*", however panellists argued that game load management was just as important for consideration as training load. Panellists agreed that injury prevention strategies are a multidisciplinary team responsibility between sports science and medical departments [44]. Rewording of this statement incorporated the use of a multidisciplinary approach with consideration to training and game load management, in addition to strength and conditioning gym programmes to achieve consensus [45].

**Statement 13:** *It is unrealistic for practitioners to use bio-banding as a method to reduce injury risk without greater training and research dissemination.*

## Supplementary information

Despite a plethora of research articles dedicated to bio-banding [22, 26], it appears to be a poorly understood concept from a practitioner perspective. From an industry perspective, the panellists suggested bio-banding was used as a talent/physical development strategy rather than an injury risk management method. Our panellists were unconvinced that bio-banding was an established industry strategy for protecting players from injury. It should be stated however, there is a lack of research evidence to support bio-banding as an injury prevention strategy, so this inference is based on practitioner and industry experience rather than research evidence per se.

**Statement 14:** *Additional training and education are required surrounding the prescription of interventions for academy players with growth-related conditions such as Severs disease or Osgood-Schlatter's.*

## Supplementary information

Panellists believed that training and education surrounding the management of players suffering from growth-related injuries and symptoms is lacking [2]. They felt that this originated from a university degree level, whereby graduate students were entering academy settings in full-time job roles, without any previous experience of dealing with these types of injuries and symptoms. This statement was amended to include examples of growth-related conditions for greater clarity. The entire statement was also changed, as it was originally assumed that sport

science staff would feel supported and would have received training on how to deal with these types of conditions however, the reality from this study is somewhat different.

**Statement 15:** *Qualified performance/sports science staff in academy environments have sufficient knowledge and expertise of taking growth-related measurements and using common maturity assessment methods in practice [e.g., Khamis-Roche., 1994; Mirwald., 2002] to determine a players' maturity status and the timing of the adolescent growth spurt.*

## Supplementary information

An emerging theme within academy environments is the responsibility of conducting maturity-related assessments being placed on unqualified sports science staff such as interns. Panellists were satisfied that qualified staff have adequate knowledge and expertise of using non-invasive methods to collect maturity-related data, and they can interpret and apply the results. However, they expressed some concerns that qualified staff are conducting these assessments less frequently and instead the responsibility is placed on staff with little or no training [46]. This statement was therefore amended to be targeted towards qualified sports science staff in academies.

**Statement 16:** *Apophysitis conditions around the hip are more difficult to diagnose than apophysitis conditions around the foot and ankle and require a specialist assessment.*

## Supplementary information

It was well accepted that apophysitis conditions around the hip are more difficult to diagnose than around the foot and ankle. No further comments were made to explain the reasoning behind it, but it demonstrates an area for future research to explore either within sports science or physiotherapy. No adjustments were made for this statement.

## General discussion

Despite the geographical and professional variability of our panellists, there were some areas that reached broad consensus. Firstly, maturity-related data collection is completed for multiple purposes, to support the long-term development of players [13]. Secondly, longitudinal monitoring is preferable to accurately assess growth patterns, with increased screening for players immediately pre/circa-PHV to implement strength and conditioning and training load strategies associated with growth and maturity-related injury factors [4, 10, 43]. Thirdly, panellists believed that the validity of maturity-related assessments could be improved with greater training/education for staff when conducting assessments and when managing players with growth-related conditions/symptoms [46].

The findings from this Delphi study suggest that panellists consider phases of accelerated growth such as PHV, muscle strength/flexibility imbalances, altered biomechanics e.g. 'adolescent awkwardness', maturity status (% predicted adult height) and the period circa-PHV and post-PHV (up to 12 months), as highly important maturity-related injury risk factors (median score ≥ 7). Fluctuations in lean body mass, lower/upper extremity growth rates and the period leading up to PHV (12 months) were perceived as less important (median score 4–6).

When investigating the complexity of assessing growth-related conditions, one interesting finding was the belief that hip apophysitis injuries are more difficult to treat and diagnose than those of the foot/ankle (median score = 7). The hip joint is exposed to a higher risk of injury, due to vigorous and repetitive muscular contractions on the musculotendinous junction and its bony attachments, commonly associated with sport-specific actions in sports such as soccer [47]. This is supported with a reported 20% prevalence of osteochondral disorders affecting the pelvis, ischium, anterior inferior iliac spine, anterior superior iliac spine, iliac crest and lesser trochanter in French academy soccer players [48]. In general, apophysitis injuries are

diagnosed based on clinical and radiographic findings [47], however, apophysitis injuries of the hip are still commonly misdiagnosed and treated as a muscular strain [49], which highlights the complexity of the hip joint, in addition to the diagnostic and treatment challenges for practitioners [50]. This would suggest that further training and education for practitioners is required, to help identify relevant symptoms and implement appropriate treatment strategies associated with these types of growth-related conditions, given its high prevalence and injury burden among academy soccer players [51].

Previous literature has proposed a variety of maturity-related injury risk factors [6–12], in conjunction with varying rates of injury incidence associated with the stages of PHV [15, 16]. The present study findings indicate that panellists believed that the period during PHV and 12 months post-PHV were more important for growth-related injury risk. Players who are circa-PHV may experience more growth-related injuries (e.g., Osgood-Schlatter's, Sever's), whilst knee/ankle muscular and articular injuries are more common post-PHV, alongside higher injury incidence which may be due to higher intensity and volume of training [15]. Imbalances between muscular strength and flexibility, coupled with altered biomechanics associated with 'adolescent awkwardness' [10, 37] were also deemed as important risk factors. Traditionally, it has been suggested that periods of accelerated growth (e.g., PHV) result in decreased muscle flexibility, further offsetting the balance between strength and flexibility, which can increase the vulnerability of the skeletal system to injury [51]. Imbalances between strength and flexibility following a period of growth has also been suggested to reduce the ability of the cartilaginous structures to cope with high-level stress, leading to overuse and apophysitis injuries [10]. Accompanied with this strength/flexibility imbalance, temporary delays in motor control are reportedly common during and after accelerated phases of growth [37]. This can lead to 'adolescent awkwardness' due to an accelerated growth of the lower extremities combined with poor neuromuscular control, which can potentially increase injury risk during this period, although it is important to note that no studies have confirmed a definitive link between 'adolescent awkwardness' and injury risk [37].

Previous research has also shown that injuries follow a growth specific pattern associated with maturity status (< 88% to > 96% predicted adult height) [15], which was also perceived as important by the panellists in this study. This demonstrates the difficulties to implement targeted injury prevention programmes within youth academies, given the variety of injuries associated with individual player maturation and growth [4]. Therefore, it is our contention that practitioners should identify and use appropriate injury prevention and training load strategies, depending on a player's stage of maturation with consideration to the area's most at risk. We also recommend that researchers work more closely with practitioners in academy environments, to implement effective ways of monitoring and assessing the maturity-related risk factors that were deemed highly important by the panellists in this study.

Recent literature has suggested that injury prevention is one highly important reason for maturity-related data collection in German youth academies (85% importance), with other important uses including load management, player recruitment and bio-banding (95%, 75% and 65% importance respectively) [13]. These findings concur with the present study, in that maturity-related data collection is completed for multiple reasons. Only 40% of panellists believed that maturity-related data collection was primarily for injury prevention, with comments such as "*Other key reasons include talent identification and development*" (Panellist 1), "*Data from maturity assessment can be utilised for several purposes, but I don't think one is a priority over others*" (Panellist 2), "*Physical staff would say injury prevention, other staff may say performance related / profiling reasons*" (Panellist 3). Collectively, the findings from this study and elsewhere [13] demonstrate that maturity data collection is completed to assist with the long-term development of a player from both physical and performance-related perspectives.

One controversial finding from the present study was the belief that maturity-related data shouldn't be used for recruitment or retain/release decisions (median score = 5). Recent literature has alluded to the importance of maturity-related data collection for recruitment (75% importance) and retain/release decisions (58% importance) in German academies [13]. However, panellists in the current study believed "*this should never be the case*" (Panellist 4) and that "*growth and maturity-related data should never be the be-all and end-all of retain/release and recruitment decisions*" (Panellist 1). This somewhat contradicts the results from previous studies and highlights the differences in culture between UK and German soccer academy practices surrounding maturity-related data collection. This study and the earlier study from Germany [13] concur that further research is needed to investigate the reasons behind the inconsistent motives for maturation assessments [13].

Regarding the pattern of maturity-related data collection, Towlson *et al.*, [10] reported that practitioners collect maturity-related data every three months, with an increased focus on players pre/circa-PHV. This is in line with the findings in this study, with panellists commenting that "*Three months seems to be a sensible timeframe to ensure regular data... If we feel that a player is about to approach, or is going through PHV, we might increase testing frequency to every 6 weeks*" (Panellist 4), coupled with a 100% agreement that the regular collection (three monthly) of maturity data can facilitate with more beneficial outcomes for youth players for long term athletic development. Furthermore, there is agreement between the findings presented here and those from a German academy study, in that maturity-related data is used to inform training load management for players at different stages of their growth and maturation [13]. Panellists commented that "*Gym programmes will be tailored more around those players with a close PHV proximity*" (Panellist 5), "*Modifications will be made to training and match loads (volume), with additional supplementary exercises given in the gym*" (Panellist 4). This suggests that individual load management and gym programmes are perceived to be the most effective injury prevention strategies, which are informed by maturity-related data, according to panellists in this study (100% agreement) and elsewhere [2].

One important consideration for improving maturity-related data collection practices is to standardise the way these assessments are conducted [10]. Collectively, panellists believed that standardisation of these assessments is important to gather more reliable growth-related data, as currently data is "*collected (with the upmost respect) by part-time physio's who have had no formal training.*" (Panellist 4), with suggestions that maturity assessments "*can be performed by interns*" *(*Panellist 6). This has implications for injury risk in youth players, given that inaccurate categorisation of a player's maturity status can have negative implications for training load management and strength and conditioning programmes for injury prevention [10]. Current practices could be influenced by staffing levels within academy systems. Recent findings from the top four leagues in Germany have shown that clubs can have less than six full or part-time staff within sports science and medicine departments [13]. The limited staffing dedicated to sports science and medicine departments demonstrates the time and logistical constraints often facing practitioners in their respective environments and can explain the increased responsibility placed on unqualified staff such as interns. It is our contention that researchers should be working more closely with practitioners to address and overcome some of the barriers they face on a daily basis, given the staff shortages that are apparent within soccer academies [10].

The use of non-invasive methods to assess maturation and growth in youth players has become common practice in academy systems [8]. The findings from this study suggest that the panellists perceived current non-invasive methods as sub-optimal for assessing maturation in youth players (70% agreement). Similarly, panellists were "*unfamiliar*" (Panellist 4) with a lot of the proposed methods for assessing maturity status and timing, apart from the Khamis-Roche [21] method (50% agreement). The Khamis-Roche [21] equation is one of the most

popular methods used to assess maturity status and timing in academy players [17]. This could imply there is a cultural element associated with the use of this method, given its popularity amongst practitioners. Nevertheless, a recent review has demonstrated that no two methods produce the same estimation of adult height, skeletal age or age at PHV, with only a moderate agreement (44–50%) for maturity status classification using different non-invasive methods [24]. The findings from this study and recent review confirm suspicions that practitioners are aware that the methods they employ to assess growth and maturation in youth players are flawed and require improvement [24], however they are obliged to use these methods, given the lack of viable alternatives. Our recommendation to practitioners is to be aware of the prediction error that accompanies each non-invasive method they choose to employ. Furthermore, to improve practice, practitioners should make a conscious effort to ensure data collection is completed as reliably as possible, preferably by qualified sports science staff with appropriate qualifications and using valid equipment [10].

Bio-banding has become increasingly popular in youth soccer academies and has been endorsed by the English Premier League as a mechanism to mitigate maturity-related selection bias [26]. By tradition, bio-banding is used for physical and technical development, whilst providing opportunities for talent identification [26], however, more recent work has suggested that it can be used as a method of maturity-related injury prevention [22]. Panellists in this study stated a wide range of uses for bio-banding, however, uncertainty surrounded its use for injury prevention: "*I believe bio-banding to have many benefits (including psychosocial) but injury prevention is not one*" (Panellist 4), "*No. I believe bio-banding is more of a method of increasing technical / tactical performance*", (Panellist 3) "*I do not think bio-banding was ever intended to be used as an injury reduction tool. More to provide variety and challenge for players in a physical and psychosocial manner*" (Panellist 4). This somewhat contradicts recent research findings, suggesting that bio-banding is used more for developing technical competencies as opposed to protecting players from injuries. However, panellists strongly believed that greater research and dissemination of findings surrounding bio-banding is needed (89% agreement).

## Implications for research

From an applied performance perspective, the findings from this Delphi study suggest maturity-related data forms part of an integrated and multidisciplinary approach, to support the long-term development of youth academy players in the UK. Contrary to previous research our panellists did not reach consensus on the use of maturity data for recruitment or retain/release purposes [13]. The methods used to gather maturity-related data remain somewhat unreliable, with practitioners aware of their limitations. Therefore, researchers can assist practitioners via the development of frameworks to advise and educate practitioners around best practice when using non-invasive, predictive equations during their maturity assessments. This can mitigate any concerns around reliability by highlighting the prediction error associated with maturity-estimated equations, alongside the implications of additional errors associated with false anthropometric measures (e.g., estimated parental height) [10]. It can also encourage better practice by ensuring that the practitioners responsible for conducting these types of assessments consider other statistical metrics associated with prediction error (e.g., coefficient of variation, inter/intra reliability, smallest meaningful change), in order to optimise their maturity assessments [10].

## Methodological considerations

An obvious limitation to this Delphi study was the Anglophile context of the panellists. We therefore recommend further Delphi studies are conducted in an international context to

remove the UK bias and to include other disciplines such as physiotherapy and psychology to expand the findings presented here. A secondary limitation was the moderate response rate (69%). Prior to the start of the study, we identified an ideal sample would be between 11–20 panellists [32]; however, the final sample was limited to ten panellists for rounds one and two, with one panellist dropping out during the final round, leaving a total sample of nine panellists for all three rounds. There was variability in the time spent in the panellist' current role (i.e., 3-months to 13-years) and this should be considered when interpreting these findings. Saying that, the lack of panellists may be mitigated by the industry experience and expertise of the panellists, after all it was our intention to produce recommendations relevant to this group of professionals.

## Conclusion

This Delphi study has identified some urgent areas for further research. Clarity around defining key language features used within this research area (e.g. 'movement efficiency', 'adolescent awkwardness') is warranted to validate these language terms and to create a homogenous approach to research within this area [24]. This study highlights that maturity-related data is collected and used to support the long-term development of players from physical and performance-related perspectives, but not for recruitment or retain/release decisions. The methods and practices employed during data collection remain questionable, with known limitations surrounding the use of the non-invasive methods used to complete maturity assessments, coupled with poor staff training and competency for conducting these assessments. Accelerated phases of growth and the 12-month period around PHV, maturity status (% predicted adult height), muscle strength/flexibility imbalances and 'adolescent awkwardness' were deemed as highly important maturity-related risk factors, with the belief that longitudinal and accurate monitoring of maturation every 6–12 weeks is needed within academy environments. Apophysitis injuries involving the hip/pelvis were deemed harder to diagnose and treat, with further training needed on how to handle and treat players with these types of conditions. How these findings impact player outcomes remain unknown, but it is clear that better education/training, dissemination of research findings and collaboration between researchers and practitioners is needed. It is hopeful that this study can act as an anchor between academic and practitioner environments to align objectives, implement effective interventions and build stronger partnerships between researchers and practitioners working with youth academy players, to ultimately produce better outcomes for their long-term development.

## Supporting information

**S1 File. Delphi poll round one synthesis.**
(DOCX)

**S2 File. Delphi poll round two background report.**
(DOCX)

**S3 File. Delphi poll round two synthesis.**
(DOCX)

**S4 File. Delphi poll round three background report.**
(DOCX)

**S5 File. Delphi poll round three synthesis.**
(DOCX)

## Acknowledgments

The research term would like to extend their sincere gratitude to all the panellists for their time, patience and expertise.

## Author Contributions

**Formal analysis:** David Johnson, David Hartley.

**Investigation:** Joseph Sullivan, Simon Roberts.

**Methodology:** Joseph Sullivan, Simon Roberts, David Johnson, David Hartley.

**Supervision:** Simon Roberts, Kevin Enright, Martin Littlewood.

**Visualization:** Joseph Sullivan.

**Writing – original draft:** Joseph Sullivan, Simon Roberts, Kevin Enright, David Johnson, David Hartley.

**Writing – review & editing:** Joseph Sullivan, Simon Roberts, Kevin Enright, David Johnson, David Hartley.

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
