## [Decision Letter · Decision Letter 0]

13 Sep 2024

PONE-D-24-23052Consensus on Maturity-Related Injury Risks and Prevention in Youth Soccer: A Delphi Study.PLOS ONE

Dear Dr. Sullivan,

Thank you for submitting your manuscript to PLOS ONE. After careful consideration, we feel that it has merit but does not fully meet PLOS ONE’s publication criteria as it currently stands. Therefore, we invite you to submit a revised version of the manuscript that addresses the points raised during the review process.

We look forward to receiving your revised manuscript.

Kind regards,

Julio Alejandro Henriques Castro da Costa

Academic Editor

PLOS ONE

Journal Requirements:

Reviewers' comments:

Reviewer's Responses to Questions

**Comments to the Author**

1. Is the manuscript technically sound, and do the data support the conclusions?

Reviewer #1: Yes

2. Has the statistical analysis been performed appropriately and rigorously? 

Reviewer #1: Yes

3. Have the authors made all data underlying the findings in their manuscript fully available?

Reviewer #1: Yes

4. Is the manuscript presented in an intelligible fashion and written in standard English?

Reviewer #1: Yes

5. Review Comments to the Author

Reviewer #1: Thank you for the opportunity to review this article. This paper seems really a good contribution to the literature with its strong sides such as seeking for a consensus. I congratulate authors for the well-structured, well-applied and strong methodology and results in particular. As stated by the authors, although the number of participants is just below the suggested numbers, there is still quality data saying a lot to the reader.

However, I have some concerns about the citations and statements in the article. I suggest authors to think twice while using the word “holistic development”. Throughout the paper, there are some places this phenomenon mentioned but I believe that it is not appropriate to do so. For example, in the sentence starting on Line 705 “This study highlights that maturity related data is collected and used holistically, to support the long-term development of players from physical, performance-related and psychosocial perspectives, but not for recruitment or retain/release decisions.”, here in this study I cannot see any concern or statements for psychosocial or emotional perspective even in participants’ notes provided as supplements. Another one might be the sentence starting on Line 521 “Firstly, maturity-related data collection is completed for multiple purposes, to support the long-term development of players from a holistic perspective (12).” I think the paper cited here with number 12 does not have any connotation for collecting data with a holistic perspective either. So, I suggest authors to think twice on the phenomenon (i.e. holistic development/perspective) and do the revisions accordingly in their paper.

Moreover, for some citations throughout the paper, I suggest authors to go to the very first paper saying/showing/presenting/giving that specific finding or suggestions. For example, in the sentence starting on Line 70 “For example, youth players of the same chronological age group can vary in biological maturity by as much as 5-6 years (2).”, here the paper cited with number 2 does not have any data regarding 5-6 years difference. So, the very first source should be found and cited here which have this finding! Again, in the sentence starting on Line 111 “These findings suggest that the non-invasive methods used to assess maturity status and timing in youth players are largely unreliable, which could have negative consequences for players’ injury risk (2)”, the same cited paper number 2, does it say so? Does it say that the degree of non-invasive methods’ predictions on maturity status and timing has negative consequences for players’ injury risk? Does this article really suggest so? Are they “largely” unreliable? Lastly, can authors check whether it is citation number 3 or number 1 suggesting the statement in the sentence starting on Line 71 “The variability in the timing of the adolescent growth spurt (occurring between 13-15 years in boys)can offer additional complexities for performance staff who wish to implement injury risk and training load management strategies (3, 4). So, again, I suggest authors to think twice for their citations.

Lastly, I suggest authors to stress more and explain why they chose Delphi design because I really think that it is important for this study and makes the article stronger.

Overall comment: I congratulate authors for their work. The manuscript seems strong with its methodology but I have some concerns with statements and citations. So, I suggest minor revision and would like to see the paper again after revisions.

6. PLOS authors have the option to publish the peer review history of their article (what does this mean?). If published, this will include your full peer review and any attached files.

Reviewer #1: No

---

## [Author Response · Author response to Decision Letter 0]

18 Sep 2024

Dear Editor, 

Thank you for completing a review of the manuscript entitled: Consensus on Maturity-Related Injury Risks and Prevention in Youth Soccer: A Delphi Study. (Manuscript ID: PONE-D-24-23052). 

My co-authors and I appreciate your time and feedback in reviewing the manuscript and are pleased you appreciate the value of the research. In accordance with some of your recommendations we have made the following changes to the manuscript and provide a detailed description of the changes we have made (see response to reviewer document). Following these revisions to the manuscript we consider it now ready for publication. 

Once again, thank you for your time and feedback. We look forward to hearing your decision. 

Yours sincerely 

The authorial team.

---

## [Decision Letter · Decision Letter 1]

30 Sep 2024

PONE-D-24-23052R1Consensus on Maturity-Related Injury Risks and Prevention in Youth Soccer: A Delphi Study.PLOS ONE

Dear Dr. Sullivan,

Thank you for submitting your manuscript to PLOS ONE. After careful consideration, we feel that it has merit but does not fully meet PLOS ONE’s publication criteria as it currently stands. Therefore, we invite you to submit a revised version of the manuscript that addresses the points raised during the review process.

We look forward to receiving your revised manuscript.

Kind regards,

Julio Alejandro Henriques Castro da Costa

Academic Editor

PLOS ONE

Journal Requirements:

Reviewers' comments:

Reviewer's Responses to Questions

**Comments to the Author**

1. If the authors have adequately addressed your comments raised in a previous round of review and you feel that this manuscript is now acceptable for publication, you may indicate that here to bypass the “Comments to the Author” section, enter your conflict of interest statement in the “Confidential to Editor” section, and submit your "Accept" recommendation.

Reviewer #1: All comments have been addressed

2. Is the manuscript technically sound, and do the data support the conclusions?

Reviewer #1: Yes

3. Has the statistical analysis been performed appropriately and rigorously? 

Reviewer #1: Yes

4. Have the authors made all data underlying the findings in their manuscript fully available?

Reviewer #1: Yes

5. Is the manuscript presented in an intelligible fashion and written in standard English?

Reviewer #1: Yes

6. Review Comments to the Author

Reviewer #1: Thank you for giving me the opportunity for reviweing this manuscript and I congradulate the authors for their great work and effort again. To me, in its current form, the manuscript looks a good contribution to the literature. I only have one concern for a citation that I have explained below. Again, congradulations to the authorial team.

Line 91 in the revised document provided to me: You have cited “Johnson, A. Monitoring the Immature Athlete. Aspetar Sports Medicine Journal. 2015; 4(1)” for your statement “For example, youth players of the same chronological age group can vary in biological maturity by as much as 5-6 years”. My question again: Does this article have empirical hard data or is it written there with referencing to another article? The answer is: Johnson (2015) did not have it either and just cited another previous paper for this statement which is “Johnson, A., Doherty, P. J., & Freemont, A. (2009). Investigation of growth, development, and factors associated with injury in elite schoolboy footballers: prospective study. BMJ (Clinical research ed.), 338, b490.”. However, Johnson et al. (2009) also used this information in their article as “In youth sport, chronological age is the usual method of dividing children into age related training and competitive groups, but between individuals in the same age group this can differ by as much as four years from skeletal age” and cited “Malina RM, Pena Reyes ME, Eisenmann JC, Horta L, Rodrigues J, MillerR. Height,mass and skeletal maturity of elite Portuguese soccer players aged 11-16 years. J Sports Sci 2000;18:685-93”. So, here we see that the original data comes from Malina et al.’s (2000) study related to that 5-6 years difference. So, please cite Malina et al.’s study instead of Johnson’s study for your statement because the hard data comes from Malina et al’s study. That would be ethical to giving their credits and dues considering the work they did.

7. PLOS authors have the option to publish the peer review history of their article (what does this mean?). If published, this will include your full peer review and any attached files.

Reviewer #1: No

---

## [Author Response · Author response to Decision Letter 1]

3 Oct 2024

Dear Editor, 

Thank you for completing a review of the manuscript entitled: Consensus on Maturity-Related Injury Risks and Prevention in Youth Soccer: A Delphi Study. (Manuscript ID: PONE-D-24-23052). 

My co-authors and I appreciate your time and feedback in reviewing the manuscript for second time and are pleased you appreciate the value of the research. In accordance with your recommendation, we have adjusted the manuscript and provide a detailed description below of the changes we have made. Following this alteration to the manuscript we consider it now ready for publication. 

Once again, thank you for your time and feedback. We look forward to hearing your decision. 

Yours sincerely 

The authorial team.

---

## [Decision Letter · Decision Letter 2]

9 Oct 2024

Consensus on Maturity-Related Injury Risks and Prevention in Youth Soccer: A Delphi Study.

PONE-D-24-23052R2

Dear Dr. Sullivan,

We’re pleased to inform you that your manuscript has been judged scientifically suitable for publication and will be formally accepted for publication once it meets all outstanding technical requirements.

Kind regards,

Julio Alejandro Henriques Castro da Costa

Academic Editor

PLOS ONE

Additional Editor Comments (optional):

Reviewers' comments:

Reviewer's Responses to Questions

**Comments to the Author**

1. If the authors have adequately addressed your comments raised in a previous round of review and you feel that this manuscript is now acceptable for publication, you may indicate that here to bypass the “Comments to the Author” section, enter your conflict of interest statement in the “Confidential to Editor” section, and submit your "Accept" recommendation.

Reviewer #1: All comments have been addressed

2. Is the manuscript technically sound, and do the data support the conclusions?

Reviewer #1: Yes

3. Has the statistical analysis been performed appropriately and rigorously? 

Reviewer #1: Yes

4. Have the authors made all data underlying the findings in their manuscript fully available?

Reviewer #1: Yes

5. Is the manuscript presented in an intelligible fashion and written in standard English?

Reviewer #1: Yes

6. Review Comments to the Author

Reviewer #1: Thank you for giving me the opportunity for reviweing this manuscript again and I congradulate the authors for their great work and effort. To me, in its current form, the manuscript looks a good contribution to the literature. Congradulations again.

7. PLOS authors have the option to publish the peer review history of their article (what does this mean?). If published, this will include your full peer review and any attached files.

Reviewer #1: No

---

## [Editor Report · Acceptance letter]

12 Oct 2024

PONE-D-24-23052R2 

PLOS ONE

Dear Dr. Sullivan, 

I'm pleased to inform you that your manuscript has been deemed suitable for publication in PLOS ONE. Congratulations! Your manuscript is now being handed over to our production team.

Kind regards, 

on behalf of

Dr. Julio Alejandro Henriques Castro da Costa 

Academic Editor

PLOS ONE
